# 3DTopia-XL: Scaling High-quality 3D Asset Generation via Primitive Diffusion

Figure 1: 3DTopia-XL generates high-quality 3D assets with smooth geometry and spatially varied textures and materials. The output asset (GLB mesh) can be seamlessly ported into graphics engines for physically-based rendering. **Recommend to open with Acrobat Reader for animation.**

## Abstract

The increasing demand for high-quality 3D assets across various industries necessitates efficient and automated 3D content creation. Despite recent advancements in 3D generative models, existing methods still face challenges with optimization speed, geometric fidelity, and the lack of assets for physically based rendering (PBR). In this paper, we introduce 3DTopia-XL, a scalable native 3D generative model designed to overcome these limitations. 3DTopia-XL leverages a novel primitive-based 3D representation, PrimX, which encodes detailed shape, albedo, and material field into a compact tensorial format, facilitating the modeling of high-resolution geometry with PBR assets. On top of the novel representation, we propose a generative framework based on Diffusion Transformer (DiT), which comprises 1) Primitive Patch Compression, 2) and Latent Primitive Diffusion. 3DTopia-XL learns to generate high-quality 3D assets from textual or visual inputs. We conduct extensive qualitative and quantitative experiments to demonstrate that 3DTopia-XL significantly outperforms existing methods in generating high-quality 3D assets with fine-grained textures and materials, efficiently bridging the quality gap between generative models and real-world applications.

## 1 Introduction

High-quality 3D assets are essential for various real-world applications like films, gaming, and virtual reality. However, creating high-quality 3D assets involves extensive manual labor and expertise. Therefore, it further fuels the demand for automatic 3D content creation techniques, which automatically generate 3D assets from visual or textual inputs by using 3D generative models.

Fortunately, rapid progress has been witnessed in the field of 3D generative models recently. Existing state-of-the-art techniques can be sorted into three categories. **1)** Methods based on Score Distillation Sampling (SDS) (Poole et al., 2022; Tang et al., 2023a) lift 2D diffusion priors into 3D representation by per-scene optimization. However, these methods suffer from time-consuming optimization, poor geometry, and multifaceted inconsistency. **2)** Methods based on sparse-view reconstruction (Hong et al., 2023; Xu et al., 2024a) that leverage large models to regress 3D assets from single- or multi-view images. Most of these methods are built upon triplane-NeRF (Chan et al., 2022) representation. However, due to the triplane's parameter inefficiency, the valid parameter space is limited to low resolutions in those models, leading to relatively low-quality 3D assets. Plus, reconstruction-based models also suffer from a low-diversity problem as deterministic methods. **3)** Methods as native 3D generative models (Yariv et al., 2023; Li et al., 2024c) aim to model the probabilistic distribution of 3D assets, generating 3D objects given input conditions. Yet, few of them are capable of generating high-quality 3D objects with Physically Based Rendering (PBR) assets, which are geometry, texture, and material packed into a GLB file.

To address the limitations above, we propose 3DTopia-XL, a high-quality native 3D generative model for 3D assets at scale. Our key idea is scaling the powerful diffusion transformer (Peebles & Xie, 2022) on top of a novel primitive-based 3D representation. At the core of 3DTopia-XL is an efficient 3D representation, PrimX, which encodes the shape, albedo, and material of a textured mesh in a compact $N \times D$ tensor, enabling the modeling of high-resolution geometry with PBR assets. In specific, we anchor $N$ primitives to the positions sampled on the mesh surface. Each primitive is a tiny voxel, parameterized by its 3D position, a global scale factor, and corresponding spatially varied payload for SDF, RGB, and material. Note that the proposed representation differentiates itself from the shape-only representation M-SDF (Yariv et al., 2023) that PrimX encodes shape, color, and material in a unified way. It also supports efficient differentiable rendering, leading to the great potential to learn from not only 3D data but also image collections. Moreover, we carefully design initialization and fine-tuning strategy that enables PrimX to be rapidly tensorized from a textured mesh (GLB file) which is ten times faster than the triplane under the same setting.

Thanks to the tensorial and compact PrimX, we scale the 3D generative modeling using latent primitive diffusion with Transformers, where we treat each 3D object as a set of primitives. In specific, the proposed 3D generation framework consists of two modules. **1)** Primitive Patch Compression uses a 3D VAE for spatial compression of each individual primitive to get latent primitive tokens; and **2)** Latent Primitive Diffusion leverages the Diffusion Transformers (DiT) (Peebles & Xie, 2022) to model global correlation of latent primitive tokens for generative modeling. Notably, the permutation equivariance of PrimX naturally supports training Transformers without positional encoding. The significant efficiency of the proposed representation allows us to achieve high-resolution generative training using a clean and unified framework without super-resolution to upscale the underlying 3D representation or post-hoc optimization-based mesh refinement.

In addition, we also carefully design algorithms for high-quality 3D PBR asset extraction from PrimX, to ensure reversible transformations between PrimX and textured mesh. An issue for most 3D generation models (Wang et al., 2024; Xu et al., 2024a) is that they use vertex coloring to represent the object's texture, leading to a significant quality drop when exporting their generation results into mesh format. Thanks to the high-quality surface modeled by Signed Distance Field (SDF) in PrimX, we propose to extract the 3D shape with zero-level contouring and sample texture and material values in a high-resolution UV space. This leads to high-quality asset extraction with considerably fewer vertices, which is also ready to be packed into GLB format for downstream tasks.

Extensive experiments are conducted both qualitatively and quantitatively to evaluate the effectiveness of our method in text-to-3D and image-to-3D tasks. Moreover, we do extensive ablation studies to motivate our design choices for a better efficiency-quality tradeoff in the context of generative modeling with PrimX. In conclusion, we summarize our contributions as follows: **1)** We propose a novel 3D representation, PrimX, for high-quality 3D content creation, which is efficient, tensorial, and renderable. **2)** We introduce a scalable generative framework, 3DTopia-XL, tailored for generating high-quality 3D assets with high-resolution geometry, texture, and materials. **3)** Practical techniques for assets extraction from 3D representation to avoid quality gap. **4)** We demonstrate the superior quality and impressive applications of 3DTopia-XL for image-to-3D and text-to-3D tasks.

## 2 RELATED WORK

**Deterministic 3D Generative Models.** Recent advancements have been focusing on deterministic reconstruction methods that regress 3D assets from single- or multi-view images. Large Reconstruction Model (LRM) (Hong et al., 2023; He & Wang, 2023) has shown that end-to-end training of a triplane-NeRF (Chan et al., 2022) regression model scales well to large datasets and can be highly generalizable. Although it can significantly accelerate generation speed, the generated 3D assets still exhibit relatively lower quality due to representation inefficiency and suffer from a low-diversity problem as a deterministic method. Subsequent works have extended this method to improve generation quality. For example, using multi-view images (Xu et al., 2024a; Li et al., 2023; Wang et al., 2023a; Siddiqui et al., 2024; Xie et al., 2024; Wang et al., 2024; Jiang et al., 2024; Boss et al., 2024) generated by 2D diffusion models as the input can effectively enhance the visual quality. However, the generative capability is actually enabled by the frontend multi-view diffusion models (Shi et al., 2023; Li et al., 2024b; Long et al., 2023) which cannot produce multi-view images with accurate 3D consistency. Another direction is to use more efficient 3D representations such as Gaussian Splatting (Kerbl et al., 2023; Tang et al., 2024; Xu et al., 2024c; Zhang et al., 2024b; Yi et al., 2024; Chen et al., 2024) and triangular mesh (Zhang et al., 2024a; Li et al., 2024a; Wei et al., 2024; Zou et al., 2023). However, few of them can generate high-quality PBR assets with sampling diversity.

**Probabilistic 3D Generative Models.** Early works on feed-forward 3D generation involves training a GAN (Goodfellow et al., 2020) from 2D image datasets (Gao et al., 2022; Chan et al., 2022; Hong et al., 2022). However, such methods fail to scale up to large-scale datasets with general 3D objects (Deitke et al., 2023b;a). Similar to 2D diffusion models for image generation, efforts have been made to train a 3D native diffusion model on conditional 3D generation. However, unlike the universal image representation in 2D, there are many different choices for 3D representations. Voxel-based methods (Müller et al., 2023) can be directly extended from 2D methods, but they are constrained by the demanding memory usage, and suffer from scaling up to high-resolution data. Point cloud based methods (Nichol et al., 2022; Nash & Williams, 2017) are memory-efficient and can adapt to large-scale datasets, but they hardly represent the watertight and solid surface of the 3D assets. Implicit representations such as triplane-NeRF offer a better balance between memory and quality (Jun & Nichol, 2023; Gupta et al., 2023; Cheng et al., 2023; Ntavelis et al., 2023; Cao et al., 2023; Chen et al., 2023a; Wang et al., 2023b; Liu et al., 2023a). There are also methods based on other representations such as meshes and primitives (Liu et al., 2023b; Yariv et al., 2023; Chen et al., 2023b; Xu et al., 2024b; Yan et al., 2024). However, these methods still struggle with generalization or producing high-quality assets. Recent methods attempt to adapt latent diffusion models to 3D (Zhang et al., 2023; Zhao et al., 2023; Zhang et al., 2024c; Wu et al., 2024; Li et al., 2024c; Lan et al., 2024; Hong et al., 2024; Tang et al., 2023b). These methods first train a 3D compression model such as a VAE to encode 3D assets into a more compact form, which allows the diffusion model to train more effectively and show strong generalization. However, they either suffer from low-resolution results or are incapable of modeling PBR materials. In this paper, we propose a new 3D latent diffusion model based on a novel representation, PrimX, which can be efficiently computed from a textured mesh and unpacked into high-resolution geometry with PBR materials.

## 3 METHODOLOGY

### 3.1 PRIMX: AN EFFICIENT REPRESENTATION FOR SHAPE, TEXTURE, AND MATERIAL

Before diving into details, we outline the following design principles for 3D representation in the context of high-quality large-scale 3D generative models: **1) Parameter-efficient**: provides a good trade-off between approximation error and parameter count; **2) Rapidly tensorizable**: can be efficiently transformed into a tensorial structure, which facilitates generative modeling with modern neural architectures; **3) Differentiably renderable**: compatible with differentiable renderer, enabling learning from both 3D and 2D data.

Given the aforementioned principles, we propose a novel primitive-based 3D representation, namely PrimX, which represents the 3D shape, texture, and material of a textured mesh as a compact $N \times D$ tensor. It can be efficiently computed from a textured mesh (typically a GLB file) and directly rendered into 2D images via a differentiable rasterizer.

Figure 2: **Illustration of PrimX.** We propose to represent the 3D shape, texture, and material of a textured mesh as a compact $N \times D$ tensor (Sec. 3.1.1). We anchor $N$ primitives to the positions sampled on the mesh surface. Each primitive $\mathcal{V}_k$ is a tiny voxel with a resolution of $a^3$, parameterized by its 3D position $\mathbf{t}_k \in \mathbb{R}^3$, a global scale factor $s_k \in \mathbb{R}^+$, and corresponding spatially varied payload $\boldsymbol{X}_k \in \mathbb{R}^{a \times a \times a \times 6}$ for SDF, RGB, and material. This tensorial representation can be rapidly computed from a textured mesh within 1.5 minutes (Sec. 3.1.2).

### 3.1.1 DEFINITION

**Preliminaries.** Given a textured 3D mesh, we denote its 3D shape as $\mathcal{S} \in \mathbb{R}^3$, where $\mathbf{x} \in \mathcal{S}$ are spatial points inside the occupancy of the shape, and $\mathbf{x} \in \partial\mathcal{S}$ are the points on the shape's boundary, *i.e.,* the shape's surface. We model the 3D shape as SDF as follows:

$$F_{\mathcal{S}}^{\mathrm{SDF}}(\mathbf{x}) = \begin{cases} -d(\mathbf{x}, \partial\mathcal{S}), & \mathbf{x} \in \mathcal{S} \\ d(\mathbf{x}, \partial\mathcal{S}), & \text{elsewise} \end{cases} \quad \text{s.t.} \ \ d(\mathbf{x}, \partial\mathcal{S}) = \min_{\mathbf{y} \in \mathcal{S}} ||\mathbf{x} - \mathbf{y}||_2. \quad (1)$$

Moreover, given the neighborhood of shape surface, $\mathcal{U}(\partial\mathcal{S}, \delta) = \{d(\mathbf{x}, \partial\mathcal{S}) < \delta\}$, the space-varied color function and material function of the target mesh are defined by:

$$F_{\mathcal{S}}^{\mathrm{RGB}}(\mathbf{x}) = \begin{cases} C(\mathbf{x}), & \mathbf{x} \in \mathcal{U} \\ 0, & \text{elsewise} \end{cases} \quad F_{\mathcal{S}}^{\mathrm{Mat}}(\mathbf{x}) = \begin{cases} \rho(\mathbf{x}), & \mathbf{x} \in \mathcal{U} \\ 0, & \text{elsewise} \end{cases}, \quad (2)$$

where $C(\mathbf{x}) : \mathbb{R}^3 \to \mathbb{R}^3$ and $\rho(\mathbf{x}) : \mathbb{R}^3 \to \mathbb{R}^2$ are corresponding texture sampling functions to get albedo and material (metallic and roughness) from UV-aligned texture maps given the 3D point $\mathbf{x}$. Eventually, all shape, texture, and material information of a 3D mesh can be parameterized by the volumetric function $F_{\mathcal{S}} = (F_{\mathcal{S}}^{\mathrm{SDF}} \oplus F_{\mathcal{S}}^{\mathrm{RGB}} \oplus F_{\mathcal{S}}^{\mathrm{Mat}}) : \mathbb{R}^3 \to \mathbb{R}^6$, where $\oplus$ denotes concatenation.

**PrimX Representation.** We aim to approximate $F_{\mathcal{S}}$ with a neural volumetric function $F_{\mathcal{V}} : \mathbb{R}^3 \to \mathbb{R}^6$ parameterized by a $N \times D$ tensor $\mathcal{V}$. For efficiency, our key insight is to define $F_{\mathcal{V}}$ as a set of $N$ volumetric primitives distributed on the surface of the mesh:

$$\mathcal{V} = \{\mathcal{V}_k\}_{k=1}^N, \text{where} \ \ \mathcal{V}_k = \{\mathbf{t}_k, s_k, \boldsymbol{X}_k\}. \quad (3)$$

Each primitive $\mathcal{V}_k$ is a tiny voxel with a resolution of $a^3$, parameterized by its 3D position $\mathbf{t}_k \in \mathbb{R}^3$, a global scale factor $s_k \in \mathbb{R}^+$, and corresponding spatially varied feature payload $\boldsymbol{X}_k \in \mathbb{R}^{a \times a \times a \times 6}$ within the voxel. Note that, the payload in PrimX could be spatially varied features with any dimensions. Our instantiation here is to use a six-channel local grid $\boldsymbol{X}_k = \{\boldsymbol{X}_k^{\mathrm{SDF}}, \boldsymbol{X}_k^{\mathrm{RGB}}, \boldsymbol{X}_k^{\mathrm{Mat}}\}$ to parameterize SDF, RGB color, and material respectively.

Inspired by Yariv et al. (2023) where mosaic voxels are globally weighted to get a smooth surface, the approximation of a textured mesh is then defined as a weighted combination of primitives:

$$F_{\mathcal{V}}(\mathbf{x}) = \sum_{k=1}^N [w_k(\mathbf{x}) \cdot \mathcal{I}(\boldsymbol{X}_k, (\mathbf{x} - \mathbf{t}_k)/s_k)], \quad (4)$$

where $\mathcal{I}(\mathcal{V}_k, \mathbf{x})$ denotes the trilinear interpolant over the voxel grid $\boldsymbol{X}_k$ at position $\mathbf{x}$. The weighting function $w_k(\mathbf{x})$ of each primitive is defined as:

$$w_k(\mathbf{x}) = \frac{\hat{w}_k(\mathbf{x})}{\sum_{j=1}^N \hat{w}_j(\mathbf{x})}, \ \ \text{s.t.} \ \ \hat{w}_k(\mathbf{x}) = \max(0, 1 - ||\frac{\mathbf{x} - \mathbf{t}_k}{s_k}||_\infty). \quad (5)$$

Once the payload of primitives is determined, we can leverage a highly efficient differentiable renderer to turn PrimX into 2D images. In specific, given a camera ray $\mathbf{r}(t) = \mathbf{o} + t\mathbf{d}$ with camera

origin $\mathbf{o}$ and ray direction $\mathbf{d}$, the corresponding pixel value $I$ is solved by the following integral:

$$I = \int_{t_{\min}}^{t_{\max}} F_{\mathcal{V}}^{\text{RGB}}(\mathbf{r}(t)) \frac{dT(t)}{dt} dt, \quad \text{s.t.} \quad T(t) = \int_{t_{\min}}^{t} \exp\left[-\left(\frac{F_{\mathcal{V}}^{\text{RGB}}(\mathbf{r}(t))}{\alpha}\right)^2\right] dt, \quad (6)$$

where we use an exponent of the SDF field to represent the opacity field. And $\alpha$ is the hyperparameter that controls the variance of the opacity field during this conversion, where we set $\alpha = 0.005$.

To wrap up, the learnable parameters of a textured 3D mesh modeled by PrimX are primitive position $\mathbf{t} \in \mathbb{R}^{N \times 3}$, primitive scale $s \in \mathbb{R}^{N \times 1}$, and voxel payload $\boldsymbol{X} \in \mathbb{R}^{N \times a^3 \times 6}$ for SDF, albedo, and material. Therefore, each textured mesh can be represented as a compact $N \times D$ tensor, where $D = 3 + 1 + a^3 \times 6$ by concatenation.

**PBR Asset Extraction.** Once PrimX is constructed, it encodes all geometry and appearance information of the target mesh within the $N \times D$ tensor. Now, we introduce our efficient algorithm to convert PrimX back into a textured mesh in GLB file format. For geometry, we can easily extract the corresponding 3D shape with Marching Cubes algorithm (Lorensen & Cline, 1998) on zero level set of $F_{\mathcal{V}}^{\text{SDF}}$. For PBR texture maps, we first perform UV unwrapping in a high-resolution UV space $(1024 \times 1024)$. Then, we get sampling points in 3D and query $\{F_{\mathcal{V}}^{\text{RGB}}, F_{\mathcal{V}}^{\text{Mat}}\}$ to get corresponding albedo and material values. Note that, we mask the UV space to get the index of valid vertices for efficient queries. Moreover, we dilate the UV texture maps and inpaint the dilated region with the nearest neighbors of existing textures, ensuring albedo and material maps smoothly blend outwards for anti-aliasing. Finally, we pack geometry, UV mapping, albedo, and material maps into a GLB file, which is ready for the graphics engine and various downstream tasks.

### 3.1.2 COMPUTING PRIMX FROM TEXTURED MESH

We introduce our efficient fitting algorithm in this section that computes PrimX from the input textured mesh in a short period of time so that it is scalable on large-scale datasets for generative modeling. Given a textured 3D mesh $F_{\mathcal{S}}$, our goal is to compute PrimX such that $F_{\mathcal{V}}(\mathbf{x}) \approx F_{\mathcal{S}}(\mathbf{x})$, s.t. $\mathbf{x} \in \mathcal{U}(\partial\mathcal{S}, \delta)$. Our key insight is that the fitting process can be efficiently achieved via a good initialization followed by lightweight finetuning.

**Initialization.** We assume all textured meshes are provided in GLB format which contains triangular meshes, texture and material maps, and corresponding UV mappings. The vertices of the target mesh are first normalized within the unit cube. To initialize the position of primitives, we first apply uniform random sampling on the mesh surface to get $\hat{N}$ candidate initial points. Then, we perform farthest point sampling on this candidate point set to get $N$ valid initial positions for all primitives. This two-step initialization of position ensures good coverage of $F_{\mathcal{V}}$ over the boundary neighborhood $\mathcal{U}$ while also keeping the high-frequency shape details as much as possible. Then, we compute the L2 distance of each primitive to its nearest neighbors, taking the corresponding value as the initial scale factor for each primitive.

To initialize the payload of primitives, we first compute candidate points in global coordinates using initialized positions $\mathbf{t}_k$ and scales $s_k$ as $\mathbf{t}_k + s_k \boldsymbol{I}$ for each primitive, where $\boldsymbol{I}$ is the unit local voxel grid with a resolution of $a^3$. To initialize the SDF value, we query the SDF function converted from the 3D shape at each candidate point, *i.e.*, $\boldsymbol{X}_k^{\text{SDF}} = F_{\mathcal{S}}^{\text{SDF}}(\mathbf{t}_k + s_k \boldsymbol{I})$. Notably, it is non-trivial to get a robust conversion from arbitrary 3D shape to volumetric SDF function. Our implementation is based on an efficient ray marching with bounding volume hierarchy that works well with non-watertight topology. To initialize the color and material values, we sample the corresponding albedo colors and material values from UV space using geometric functions $F_{\mathcal{S}}^{\text{RGB}}$ and $F_{\mathcal{S}}^{\text{Mat}}$. In specific, we compute the closest face and corresponding barycentric coordinates for each candidate point on 3D mesh, then interpolate the UV coordinates and sample from the texture maps to get the value.

**Finetuning.** Even if the initialization above offers a fairly good estimate of $F_{\mathcal{S}}$, a rapid finetuning process can further decrease the approximation error via gradient descent. Specifically, we optimize the well-initialized PrimX with a regression-based loss on SDF, albedo, and material values:

$$\mathcal{L}(\mathbf{x}; \mathcal{V}) = \lambda_{\text{SDF}}||F_{\mathcal{S}}^{\text{SDF}}(\mathbf{x}) - F_{\mathcal{V}}^{\text{SDF}}(\mathbf{x})||_1 + \lambda(||F_{\mathcal{S}}^{\text{RGB}}(\mathbf{x}) - F_{\mathcal{V}}^{\text{RGB}}(\mathbf{x})||_1 + ||F_{\mathcal{S}}^{\text{Mat}}(\mathbf{x}) - F_{\mathcal{V}}^{\text{Mat}}(\mathbf{x})||_1), \quad (7)$$

where $\forall \mathbf{x} \in \mathcal{U}$, and $\lambda_{\text{SDF}}, \lambda$ are loss weights. We employ a two-stage finetuning strategy where we optimize with $\lambda_{\text{SDF}} = 10$ and $\lambda = 0$ for the first 1k iterations and $\lambda_{\text{SDF}} = 0$ and $\lambda = 1$ for the second 1k iterations. More details are provided in Sec. A.2.4 of the supplementary document.

Figure 3: **Overview of 3DTopia-XL.** As a native 3D diffusion model, 3DTopia-XL is built upon a novel 3D representation PrimX (Sec. 3.1). This compact and expressive representation encodes the shape, texture, and material of a textured mesh efficiently, which allows modeling high-resolution geometry with PBR assets. Furthermore, this tensorial representation facilitates our patch-based compression using primitive patch VAE (Sec. 3.2). We then use our novel latent primitive diffusion (Sec. 3.3) for 3D generative modeling, which operates the diffusion and denoising process on the set of latent PrimX, naturally compatible with Transformer-based neural architectures.

## 3.2 PRIMITIVE PATCH COMPRESSION

In this section, we introduce our patch-based compression on individual primitives for two main purposes: **1)** incorporating inter-channel correlations between geometry, color, and materials; and **2)** compressing 3D primitives to latent tokens for efficient latent generative modeling.

We opt for using a variational autoencoder (Kingma, 2013) (VAE) operating on local voxel patches which compresses the payload of each primitive into latent tokens, *i.e.*, $F_{\mathrm{ae}} : \mathbb{R}^D \to \mathbb{R}^d$. Specifically, the autoencoder $F_{\mathrm{ae}}$ consists of an encoder $E$ and a decoder $D$ building with 3D convolutional layers. The encoder $F_{\mathrm{ae}}$ has a downsampling rate of $48$ that compresses the voxel payload $\boldsymbol{X}_k \in \mathbb{R}^{a^3 \times 6}$ into the voxel latent $\hat{\boldsymbol{X}}_k \in \mathbb{R}^{(a/2)^3 \times 1}$. We train this VAE with reconstruction loss:

$$\mathcal{L}_{\mathrm{ae}}(\boldsymbol{X}; E, D) = \frac{1}{N} \sum_{k=1}^{N} [||\boldsymbol{X}_k - D(E(\boldsymbol{X}_k))||_2 + \lambda_{\mathrm{kl}} \mathcal{L}_{\mathrm{kl}}(\boldsymbol{X}_k, E)], \quad (8)$$

where $\lambda_{\mathrm{kl}}$ is the weight for KL regularization over the latent space. Note that, unlike other works on 2D/3D latent diffusion models (Zhang et al., 2024c; Rombach et al., 2022) that perform global compression over all patches, our VAE only compresses each local primitive patch independently and defers the modeling of global semantics and inter-patch correlation to the diffusion model. Once the VAE is trained, we can compress the raw PrimX as $\mathcal{V}_k = \{\mathbf{t}_k, s_k, E(\boldsymbol{X}_k)\}$. It leads to a low-dimensional parameter space for the diffusion model as $\mathcal{V} \in \mathbb{R}^{N \times d}$, where $d = 3 + 1 + (a/2)^3$. In practice, this compact parameter space significantly allows more model parameters given a fixed computational budget, which is the key to scaling up 3D generative models in high resolution.

## 3.3 LATENT PRIMITIVE DIFFUSION

On top of PrimX (Sec. 3.1) and the corresponding VAE (Sec. 3.2), the problem of 3D object generation is then converted to learning the distribution $p(\mathcal{V})$ over large-scale datasets. Our goal is to train a diffusion model (Ho et al., 2020) that takes as input random noise $\mathcal{V}^T$ and conditions $\mathbf{c}$, and predicts PrimX samples. Note that, the target space for denoising is $\mathcal{V}^T \in \mathbb{R}^{N \times d}$, where $d = 3 + 1 + (a/2)^3$.

In specific, the diffusion model learns to denoise $\mathcal{V}^T \sim \mathcal{N}(\mathbf{0}, \mathbf{I})$ through denoising steps $\{\mathcal{V}^{T-1}, \ldots, \mathcal{V}^0\}$ given conditional signal $\mathbf{c}$. As a set of primitives, PrimX is naturally compatible with Transformer-based architectures, where we treat each primitive as a token. Moreover, the permutation equivariance of PrimX removes the need for any positional encoding in Transformers.

Our largest latent primitive diffusion model $g_\Phi$ is a 28-layer transformer, with cross-attention layers to incorporate conditional signals, self-attention layers for modeling inter-primitive correlations, and adaptive layer normalization to inject timestep conditions. The model $g_\Phi$ learns to predict at timestep $t$ given input condition signal:

$$g_\Phi(\mathcal{V}^t, t, \mathbf{c}) = \{\mathrm{AdaLN}[\mathrm{SelfAttn}(\mathrm{CrossAttn}(\mathcal{V}^t, \mathbf{c}, \mathbf{c})), t]\}^{28}, \quad (9)$$

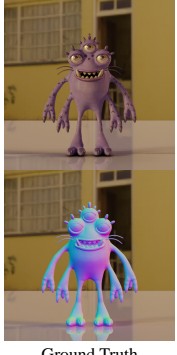

| MLP | MLP w/ PE | Triplane | Dense Voxels | **PrimX** | Ground Truth |

Figure 4: **Evaluations of different 3D representations.** We evaluate the effectiveness of different representations in fitting the ground truth's shape, texture, and material (right). All representations are constrained to a budget of 1.05M parameters. PrimX achieves the highest fidelity in terms of geometry and appearance with significant strength in runtime efficiency (Table 1) at the same time.

Table 1: **Quantitative evaluations of different 3D representations.** We evaluate the approximation error of different representations for shape, texture, and material. All representations adhere to a parameter budget of 1.05M. PrimX shows the best fitting quality, especially for the geometry (also shown in Figure 4), while having the most speedy fitting runtime. The top three techniques are highlighted in red, orange, and yellow, respectively.

| Representation | Runtime | CD $\times 10^{-4}$ $\downarrow$ | PSNR-$F_S^{\mathrm{SDF}}$ $\uparrow$ | PSNR-$F_S^{\mathrm{RGB}}$ $\uparrow$ | PSNR-$F_S^{\mathrm{Mat}}$ $\uparrow$ |
|---|---|---|---|---|---|
| MLP | 14 min | 4.502 | 40.73 | 21.19 | 13.99 |
| MLP w/ PE | 14 min | 4.638 | 40.82 | 21.78 | 12.75 |
| Triplane | 16 min | 9.678 | 39.88 | 18.28 | 16.46 |
| Dense Voxels | 10 min | 7.012 | 41.70 | 20.01 | 15.98 |
| **PrimX** | 1.5 min | 1.310 | 41.74 | 21.86 | 16.50 |

where $\mathrm{CrossAttn}(\mathbf{q}, \mathbf{k}, \mathbf{v})$ denotes the cross-attention layer with query, key, and value as input. $\mathrm{SelfAttn}(\cdot)$ denotes the self-attention layer. $\mathrm{AdaLN}(\cdot, t)$ denotes adaptive layer normalization layers to inject timestep conditioned modulation to cross-attention, self-attention, and feed-forward layers. Moreover, we employ the pre-normalization scheme (Xiong et al., 2020) for training stability. For noise scheduling, we use discrete 1,000 noise steps with a cosine scheduler during training. We opt for "v-prediction" (Salimans & Ho, 2022) with Classifier-Free Guidance (CFG) (Ho & Salimans, 2022) as the training objective for better conditional generation quality and faster convergence:

$$\mathcal{L}_{\mathrm{diff}}(\Phi) = \mathbb{E}_{t \sim [1,T], \mathcal{V}^0, \mathcal{V}^t}[|||(\sqrt{\bar{\alpha}_t}\epsilon - \sqrt{1 - \bar{\alpha}_t}\mathcal{V}^0) - g_\Phi(\mathcal{V}^t, t, \bar{\mathbf{c}}(b))||_2^2], \tag{10}$$

where $\epsilon$ is the noise sampled from Gaussian distribution, $\bar{\alpha}_t = \prod_{i=0}^{t}(1 - \beta_i)$ and $\beta_t$ comes from our cosine beta scheduler. And $b \sim \mathcal{B}(p_0)$ is a random variable sampled from Bernoulli distribution taking 0, 1 with probability $p_0$ and $1 - p_0$ respectively. And the condition signal under CFG is defined as $\bar{\mathbf{c}}(b) = b \cdot \mathbf{c} + (1 - b) \cdot \varnothing$, where $\varnothing$ is the learnable embedding for unconditional generation.

# 4 EXPERIMENTS

**Implementation Details.** We train our model on a curated subset of Objaverse (Deitke et al., 2023b) with 256k objects. Our single-view image-conditioned model utilizes DINOv2 (Oquab et al., 2023) as the conditioner, and our text-conditioned model leverages the text encoder of CLIP (Radford et al., 2021) as the conditioner. Due to the page limits, we defer more details about hyperparameters, captions, training, and inference to the supplementary document (Sec. A.2).

## 4.1 REPRESENTATION EVALUATION

**Evaluation Protocol.** We first evaluate different designs of 3D representations in the context of 3D generative modeling. Our evaluation principles focus on two aspects: **1)** runtime from GLB mesh to the representation, and **2)** approximation error for shape, texture, and material given a fixed computational budget. Given 30 GLB meshes randomly sampled from our training dataset, we take the average fitting time till convergence as runtime, which is measured as the wall time on an A100

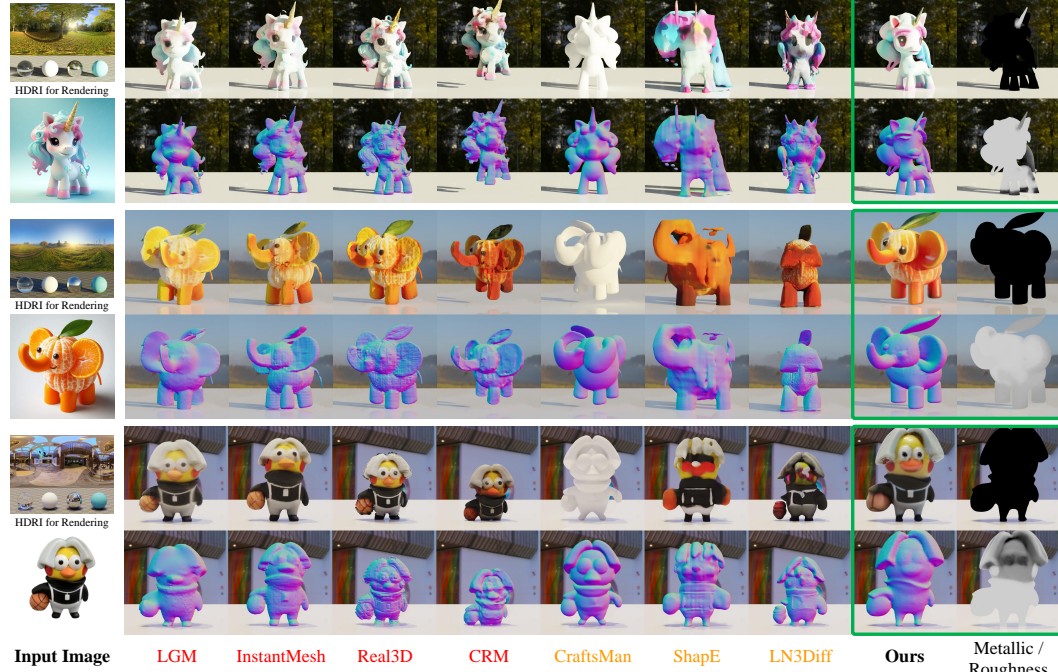

Figure 5: **Image-to-3D comparisons**. For each method, we take the textured mesh predicted from the input image into Blender and render it with the target environment map. We compare our single-view conditioned model with sparse-view reconstruction models and image-conditioned diffusion models. 3DTopia-XL achieves the best visual and geometry quality among all methods. Thanks to our capability to generate spatially varied PBR assets shown on the rightmost, our generated mesh can also produce vivid reflectance with specular highlights and glossiness.

GPU. For geometry quality, we evaluate the Chamfer Distance (CD) between the ground truth mesh and extracted mesh after the fitting and the Peak Signal-to-Noise Ratio (PSNR) of SDF values of 500k points sampled near the shape surface. For appearance quality, we evaluate the PSNR of RGB (albedo) and materials values of 500k points sampled near the surface.

**Baselines.** Given our final hyperparameters of PrimX, where $N = 2048, a = 8$, we fix the number of parameters of all representations to $2048 \times 8^3 \approx 1.05$M for comparisons. We compare four alternative representations: **1) MLP**: a pure Multi-Layer Perceptron with 3 layers and 1024 hidden dimensions; **2) MLP w/ PE**: the MLP baseline with Positional Encoding (PE) (Mildenhall et al., 2020) to the input coordinates; **3) Triplane** (Chan et al., 2022): three orthogonal 2D planes with a resolution of $128 \times 128$ and 16 channels, followed by a two-layer MLP decoder with 512 hidden dimensions. **4) Dense Voxels**: a dense 3D voxel with a resolution of $100 \times 100 \times 100$. All methods are trained with the same objectives (Eq. 7) and points sampling strategy as ours.

**Results.** Quantitative results are presented in Table 1, which shows that PrimX achieves the least approximation error among all methods, especially for geometry (indicated by CD). Besides the best quality, the proposed representation demonstrates significant efficiency in terms of runtime with nearly 7 times faster convergence speed compared with the second best, making it scalable on large-scale datasets. Figure 4 shows qualitative comparisons. MLP-based implicit methods appear to have periodic artifacts, especially for the geometry. Triplane and dense voxels yield bumpy surfaces as well as grid artifacts around the shape surface. Instead, PrimX produces the best quality with smooth geometry and fine-grained details like the thin and tapering beard.

## 4.2 IMAGE-TO-3D GENERATION

**Comparison Methods.** We run evaluations against two types of methods: **1) sparse-view reconstruction models**, and **2) image-conditioned diffusion models**. The reconstruction-based models, like LGM (Tang et al., 2024), InstantMesh (Xu et al., 2024a), Real3D (Jiang et al., 2024), CRM (Wang et al., 2024), are deterministic methods that learn to reconstruct 3D objects given four or six input views. They enable single-view to 3D synthesis by leveraging pretrained diffusion models (Shi et al., 2023; Li et al., 2024b) to generate multiple views from the input single

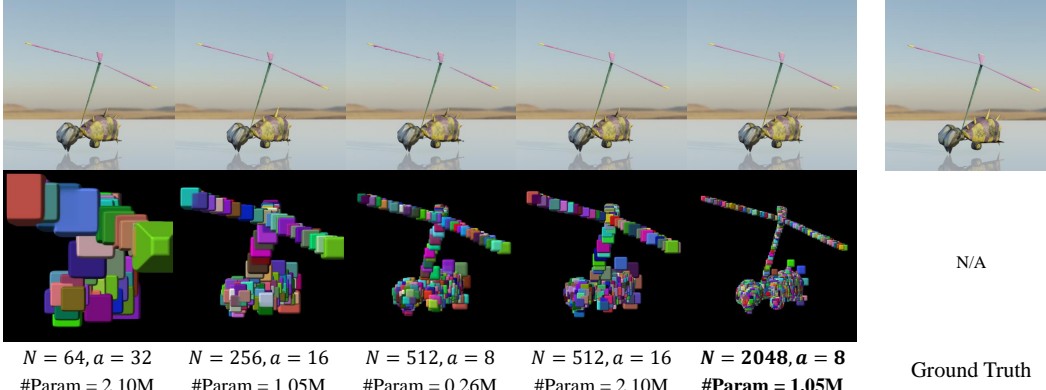

|  |  |  |  |  |  |
|---|---|---|---|---|---|
| $N = 64, a = 32$ | $N = 256, a = 16$ | $N = 512, a = 8$ | $N = 512, a = 16$ | $N = \mathbf{2048}, a = \mathbf{8}$ | Ground Truth |
| #Param = 2.10M | #Param = 1.05M | #Param = 0.26M | #Param = 2.10M | **#Param = 1.05M** |  |

Figure 6: **Ablation studies of the number and resolution of primitives.** Our final setting ($N = 2048, a = 8$) has the optimal approximation quality of ground truth, especially for fine-grained details like thin rotor blades. We visualize the corresponding PrimX at the bottom.

Table 2: **Quantitative analysis of the number ($N$) and resolution ($a$) of primitives.**

| # Primitives | Resolution | # Parameters | PSNR-$F_{\mathcal{S}}^{\mathrm{SDF}}$ ↑ | PSNR-$F_{\mathcal{S}}^{\mathrm{RGB}}$ ↑ | PSNR-$F_{\mathcal{S}}^{\mathrm{Mat}}$ ↑ |
|---|---|---|---|---|---|
| $N = 64$ | $a^3 = 32^3$ | 2.10M | 61.05 | 22.18 | 18.10 |
| $N = 256$ | $a^3 = 16^3$ | 1.05M | 59.05 | 23.50 | 18.61 |
| $N = 512$ | $a^3 = 8^3$ | 0.26M | 59.57 | 22.58 | 18.50 |
| $N = 512$ | $a^3 = 16^3$ | 2.10M | 62.89 | 23.92 | 18.21 |
| $N = 2048$ | $a^3 = 8^3$ | 1.05M | 62.52 | 24.23 | 18.53 |

image. However, as methods for reconstruction heavily rely on the input multi-view images, those methods suffer from multi-view inconsistency caused by the frontend 2D diffusion models. The feed-forward diffusion models, like CraftsMan (Li et al., 2024c), Shap-E (Jun & Nichol, 2023), LN3Diff (Lan et al., 2024), are probabilistic methods that learn to generate 3D objects given input image conditions. All methods above only model the shape and color without considering roughness and metallic while our method is suitable to produce those assets.

**Results.** Figure 5 demonstrates qualitative results. To fairly compare the capability of generating 3D assets ready for rendering, we take the exported textured mesh from each method into Blender (Community, 2018) and render it with the target environment map. For methods that cannot produce PBR materials, we assign the default diffuse material. Existing reconstruction-based models fail to produce good results which may suffer from multiview inconsistency and incapability to support spatially varied materials. Moreover, these reconstruction models are built upon triplane representation which is not parameter-efficient. This downside limits the spatial resolution of the underlying 3D representation, leading to the bumpy surface indicated by the rendered normal. On the other hand, existing 3D diffusion models fail to generate objects that are visually aligned with the input condition. While CraftsMan is the only method that has comparable surface quality as ours, they are only capable of generating 3D shapes without textures and materials. In contrast, 3DTopia-XL achieves the best visual and geometry quality among all methods. Thanks to our capability to generate spatially varied PBR assets (metallic/roughness), our generated mesh can also produce vivid reflectance with specular highlights even under harsh environmental illuminations. We also conduct a **user study** in the form of an output evaluation (Bylinskii et al., 2022), where our method performs the best. Please refer to the supplementary (Sec. A.3.1) for detailed setup and results.

## 4.3 TEXT-TO-3D GENERATION

Note that, as a pure diffusion model, our text-driven generation is done by direct textual conditioning, without relying on complicated text-to-multiview followed by reconstruction models. We conduct quantitative evaluations against native text-to-3D generative models. Given a set of unseen text prompts, we take the CLIP Score as the evaluation metric which is the cosine similarity between the text embedding and image embedding in the joint text-image space of the CLIP model (Radford et al., 2021). We take the front-view rendering from each method to compute the image embedding. We mainly compare two methods with open-source implementations: Shap-E (Jun & Nichol, 2023) and 3DTopia (Hong et al., 2024). Shap-E directly generates implicit functions of 3D objects condi-

Table 3: **Analysis of different compression rates for VAE.** $f$ stands for the compression rate between input and latent.

| # Primitives | VAE input | Latent | $f$ | PSNR ↑ |
|---|---|---|---|---|
| $N = 256$ | $6 \times 16^3$ | $6 \times 4^3$ | 64 | 22.92 |
| $N = 256$ | $6 \times 16^3$ | $1 \times 4^3$ | 384 | 19.80 |
| $N = 256$ | $6 \times 16^3$ | $1 \times 8^3$ | 48 | 23.33 |
| $N = 2048$ | $6 \times 8^3$ | $1 \times 2^3$ | 384 | 18.48 |
| $N = 2048$ | $6 \times 8^3$ | $1 \times 4^3$ | 48 | 24.51 |

Table 4: **Text-to-3D Evaluations.** We evaluate the CLIP Score between input prompts and front-view renderings of output 3D assets.

| Methods | CLIP Score ↑ |
|---|---|
| ShapE | 21.98 |
| 3DTopia | 22.54 |
| **Ours** | 24.33 |

tioned on texts. 3DTopia adopts a hybrid 2D and 3D diffusion prior by using feedforward triplane diffusion followed by optimization-based refinement. As shown in Table 4, our method achieves better alignment between input text and rendering of the generated asset. We defer the qualitative results in the supplementary (Sec. A.3.5) due to the space limit.

## 4.4 FURTHER ANALYSIS

**Number and Resolution of Primitives.** As a structured and serialized 3D representation, the number of primitives $N$ and the resolution of each primitive $a$ are two critical factors for the efficiency-quality tradeoff in PrimX. More and larger primitives often lead to better approximation quality. However, it results in a longer set length and deeper feature dimensions, causing inefficient long-context attention computation and training difficulty of the diffusion model. Therefore, we explore the impact of the number and resolution of primitives on different parameter budgets. We evaluate the PSNR of SDF, albedo, and material values given 500k points sampled near the surface. As shown in Table 2, given a fixed parameter count, a larger set of primitives appears to have a better approximation of SDF, texture, and material. Moreover, increasing the resolution of each primitive can reduce the approximation error. However, its benefit is marginal as the number of primitives is enough. The visualization in Figure 6 also confirms this observation. The alternative with ($N = 64, a = 32$) produces poor geometry even with more parameter count since larger local primitives have higher chances to waste parameters in empty space. Furthermore, a longer sequence will increase the GFlops of DiT which also leads to better generation quality (Table 5). Therefore, we tend to use a large set of primitives with a relatively small local resolution.

**Patch Compression Rate.** The compression rate of our primitive patch-based VAE (Sec. 3.2) is also an important design choice. Overall, as a patch-based compression, we aim to do spatial compression to save computation instead of global semantic compression (Rombach et al., 2022). Empirically, a higher compression rate leads to a more efficient latent diffusion model with larger batch sizes or model sizes. On the contrary, extreme compression often accompanies loss of information. Therefore, we analyze different compression rates given two different set lengths $N = 256$ and $N = 2048$ with the same parameter count of PrimX. For the evaluation metric, we compute the PSNR between the VAE's output and input on 1k random samples from the dataset to measure its reconstruction quality. Table 3 shows the results where the final choice of $N = 2048$ with compression rate $f = 48$ achieves the optimal VAE reconstruction. The setting with $N = 256, f = 48$ has the same compression rate but lower reconstruction quality and a latent space with higher resolution, which we find difficulty in the convergence of the latent primitive diffusion model $g_\Phi$.

Besides the ablation studies above, we also analyze **1) the model scaling**, **2) the sampling diversity**, and **3) PrimX initialization** of 3DTopia-XL, which are deferred to the supplementary (Sec. A.3).

## 5 DISCUSSION

We present 3DTopia-XL, a native 3D diffusion model for PBR asset generation given textual or visual inputs. Central to our approach is PrimX, an innovative primitive-based 3D representation that is parameter-efficient, tensorial, and renderable. It encodes shape, albedo, and material into a compact $N \times D$ tensor, enabling the modeling of high-resolution geometry with PBR assets. On top of PrimX, we propose Latent Primitive Diffusion for scalable 3D generative models, together with practical techniques to export PBR assets ready for graphics pipelines. Extensive evaluations demonstrate the superiority of 3DTopia-XL in text-to-3D and image-to-3D tasks, showing its great potential for 3D generative foundation models.

## 6 ETHICS STATEMENT

The main focus of 3DTopia-XL is offering an automatic framework for layman users without 3D modeling expertise to create 3D mesh with PBR assets, which are ready to use in the industrial pipeline. This increased accessibility of 3D modeling tools enabled by our generative models might be misused to create 3D content that is misleading or be misused to provide assets for fake media.

## 7 REPRODUCIBILITY STATEMENT

We have thoroughly introduced our method in Sec. 3 and provided implementation details in the supplementary material (Sec. A.2), which ensures reproducibility. Furthermore, we will release the source code and pretrained model weights upon the paper's acceptance.

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

# A APPENDIX

This supplementary material is organized as follows:

- Sec. A.1 provides further discussions, including the main difference between PrimX and existing 3D representations (Sec. A.1.1) and limitations (Sec. A.1.2).
- Sec. A.2 documents the implementations details of 3DTopia-XL, including dataset and PrimX hyperparameters (Sec. A.2.1), conditioner and captions (Sec. A.2.2), model details and hyperparameters (Sec. A.2.3), and algorithms of reversible conversion between PrimX and mesh (Sec. A.2.4).
- Sec. A.3 introduces further experiments and evaluations, including user study (Sec. A.3.1), model scaling (Sec. A.3.2), sampling diversity (Sec. A.3.3), additional ablation studies (Sec. A.3.4), and more qualitative results (Sec. A.3.5).
- Besides, we also attach a **demo video** to demonstrate the key idea and qualitative results.

## A.1 DISCUSSION

### A.1.1 DIFFERENCE WITH RELATED WORK

The core of our work is the proposed novel 3D representation, PrimX, that can model high-quality 3D shape, texture, and material in a unified and tensorial representation. It is worth highlighting the advantages of PrimX compared with other 3D representations in the **generative context**.

**PrimX v.s. Implicit Vector Set.** Previous works (Zhang et al., 2023; 2024c) introduce the implicit vector set to encode a 3D shape globally. PrimX differentiates itself from the implicit vector set in three aspects:

- PrimX encodes not only shape but also texture and material in a unified way, which removes the necessity for a two-stage framework that generates shape and texture separately.
- PrimX is differentiable renderable while implicit vector set can be only exported to meshes.
- PrimX is explicit and explainable for each token feature which facilitates 1) data augmentation by applying color transformation similar to (Karras et al., 2020); and 2) downstream tasks like inpainting by explicitly masking certain tokens.

**PrimX v.s. M-SDF (Yariv et al., 2023).** M-SDF introduces a shape-only representation to encode SDF of 3D mesh into mosaic voxels. PrimX has two distinct differences compared to M-SDF:

- M-SDF only represents 3D shape, while our method finds a unified way to encode shape, texture, and material with high quality.
- M-SDF is specialized to 3D domain while our representation can be differentiably rendered into 2D images.

**PrimX v.s. 3DGS (Kerbl et al., 2023).** As a trending representation for 3D reconstruction, 3DGS is known for its efficiency as a primitive-based volumetric representation. However, the number of Gaussians required to represent a high-quality 3D object is considerably high (hundreds of thousands) compared with PrimX (N=2048). This long context property will lead to training difficulty and inefficient attention computation in the generative context where the set of Gaussians is operated by DiT (Peebles & Xie, 2022). Instead, PrimX can be treated as an "interpolation" between fully point-based representation (3DGS) and fully voxel-based representation (dense voxel) that groups primitives into explicit structured local voxels. This hybrid operation significantly reduces the number of primitives, leading to a shorter context that boosts the training of the Transformer.

### A.1.2 LIMITATIONS AND FUTURE WORK

It is important to note that 3DTopia-XL has been trained on a considerably large-scale dataset. However, there is still room for improvement in terms of quality. Different from existing high-quality 3D diffusion models (Yariv et al., 2023; Zhang et al., 2024c) which operate on 3D representations

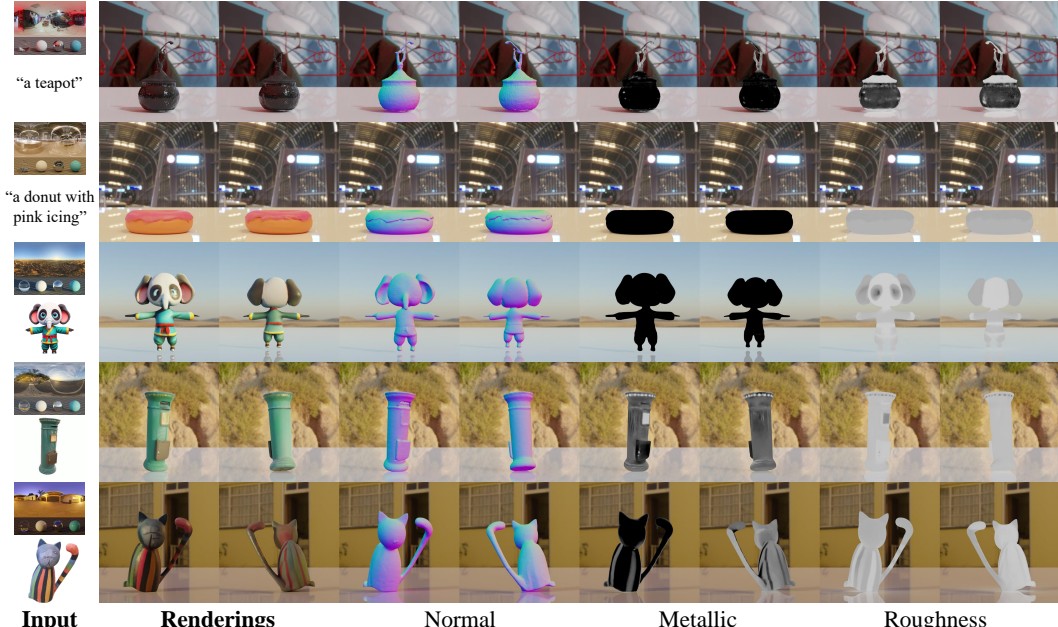

Figure 7: **3DTopia-XL can generate 3D assets directly from texts or single-view images.** We present text-conditioned generation in the top two rows and image-conditioned generation in the bottom three rows.

that are not differentiably renderable, 3DTopia-XL maintains the ability to directly learn from 2D image collections thanks to PrimX's capability of differentiable rendering (Eq. 6). This opens up new opportunities to learn 3D generative models from a mixture of 3D and 2D data, which can be a solution to the lack of high-quality 3D data. Moreover, as an explicit representation, PrimX is interpretable and easy to drive. By manipulating primitives or groups of primitives, it is also fruitful to explore dynamic object generation and generative editing.

## A.2 IMPLEMENTATION DETAILS

### A.2.1 DATA STANDARDIZATION

**Datasets.** The scale and quality of 3D data determine the quality and effectiveness of 3D generative models at scales. We filter out low-quality meshes, such as fragmented shapes and large-scale scenes, resulting in a refined collection of 256k objects from Objaverse (Deitke et al., 2023b). Computing PrimX on large-scale datasets involves two critical steps: **1)** Instantiation of sampling functions $\{F_{\mathcal{S}}^{\mathrm{SDF}}, F_{\mathcal{S}}^{\mathrm{RGB}}, F_{\mathcal{S}}^{\mathrm{Mat}}\}$ from a GLB file and **2)** Execution of the fitting algorithm in Sec. 3.1.2. Given the massive amount of meshes from diverse sources in Objaverse, there are challenges for properly instantiating the sampling functions in a universal way such as fragmented meshes, non-watertight shapes, and inconsistent UVs. Our standardized procedure starts with loading the GLB file as a connected graph. We filter out subcomponents that have less than 3 face adjacency which typically represent isolated planes or grounds. After that, all mesh subcomponents are globally normalized to the unit cube $[-1, 1]$ given one unique global bounding box. Then, we instantiate geometric sampling functions for each mesh subcomponent for SDF, texture, and material values.

**PrimX Hyperparameters.** To get a tradeoff between computational complexity and approximation error, we choose our PrimX to have $N = 2048$ primitives where each primitive's payload has a resolution of $a = 8$. It indicates that the sequence length of our primitive diffusion Transformer is also 2048 where each token has a dimension of $d = 3 + 1 + (a/2)^3 = 68$. For the rapid fine-tuning stage for computing PrimX, we sample 500k points from the target mesh, where 300k points are sampled on the surface and 200k points are sampled with a standard deviation of 0.01 near

the surface. The finetuning stage is run for 2k iterations with a batch size of 16k points using an Adam (Kingma & Ba, 2014) optimizer at a learning rate of $1 \times 10^{-4}$.

### A.2.2 CONDITION SIGNALS

**Conditioners.** The conditional generation formulation in Sec. 3.3 is compatible with most modalities. In this paper, we mainly explored conditional generation on two modalities, images and texts. For image-conditioned models, we leverage pretrained DINOv2 model (Oquab et al., 2023), specifically "DINOv2-ViT-B/14"[1], to extract visual tokens from input images (at a resolution of $518 \times 518$) and take it as the input condition $\mathbf{c}$. For text-conditioned models, we leverage the text encoder of the pretrained image-language model (Radford et al., 2021), namely "CLIP-ViT-L/14"[2], to extract language tokens from input texts.

**Images.** Thanks to our high-quality representation PrimX and its capability for efficient rendering, we do not need to undergo the complex and expensive rendering process like other works (Hong et al., 2023), which renders all raw meshes into 2D images for training. Instead, we opt to use the front-view image rendered by Eq. 6 which is **1)** efficient enough to compute on-the-fly, and **2)** consistent with the underlying representation compared with the rendering from the raw mesh.

**Text Captions.** We use 200,000 samples from Objaverse to generate text captions. For each object, six different views are rendered against a white background. We then use GPT-4V to generate keywords based on these images, focusing on aspects such as geometry, texture, and style. While we pre-define certain keywords for each aspect, the model is also encouraged to generate more context-specific keywords. Once the keywords are obtained, GPT-4 is employed to summarize them into a single sentence, beginning with 'A 3D model of...'. These text captions are subsequently prepared as input conditions.

---

**Algorithm 1:** Computing PrimX from a Textured Mesh (GLB format)

**Input** : GLB mesh $F_{\mathcal{S}}$, number of primitives $N$, voxel resolution $a$, number of candidates $\hat{N}$

▷ *Initialization*

$F_{\mathcal{S}} \leftarrow (F_{\mathcal{S}}^{\text{SDF}} \oplus F_{\mathcal{S}}^{\text{RGB}} \oplus F_{\mathcal{S}}^{\text{Mat}})$        ▷ parse volumetric sampling functions

$\{\hat{\mathbf{t}}_k\}_{k \in [\hat{N}]} \leftarrow$ uniform random sampling of $\partial \mathcal{S}$

$\{\mathbf{t}_k\}_{k \in [N]} \leftarrow$ farthest point sampling of $\{\hat{\mathbf{t}}_k\}_{k \in [\hat{N}]}$

**for** $i \leftarrow 1$ **to** $N$ **do**
    $s_i \leftarrow$ L2 distance to its nearest neighbors in $\{\mathbf{t}_k\}_{k \in [N]}$
    $\boldsymbol{X}_i^{\text{SDF}} \leftarrow F_{\mathcal{S}}^{\text{SDF}}(\mathbf{t}_i + s_i \boldsymbol{I})$        ▷ $\boldsymbol{I}$ is the local voxel grid
    $\mathbf{t}_i^{\text{uv}} \leftarrow$ UV and barycentric coordinates of the nearest face for $(\mathbf{t}_i + s_i \boldsymbol{I})$
    $\boldsymbol{X}_i^{\text{RGB}} \leftarrow F_{\mathcal{S}}^{\text{RGB}}(\mathbf{t}_i^{\text{uv}})$
    $\boldsymbol{X}_i^{\text{Mat}} \leftarrow F_{\mathcal{S}}^{\text{Mat}}(\mathbf{t}_i^{\text{uv}})$
    $\boldsymbol{X}_i \leftarrow (\boldsymbol{X}_i^{\text{SDF}} \oplus \boldsymbol{X}_i^{\text{RGB}} \oplus \boldsymbol{X}_i^{\text{Mat}})$        ▷ $\oplus$ denotes concatenation
    $\mathcal{V}_i \leftarrow \{\mathbf{t}_i, s_i, \boldsymbol{X}_i\}$

$\mathcal{V} \leftarrow \{\mathcal{V}_k\}_{k \in [N]}$

▷ *Rapid Finetuning*

**while** *not converged* **do**
    $\{\mathbf{x}_i\}_{i \in [B]} \leftarrow$ random sampling of $\mathcal{U}(\partial \mathcal{S}, \delta)$ with a batch size of $B$
    Take a gradient descent step with $\nabla_{\mathcal{V}} \mathcal{L}(\mathbf{x}; \mathcal{V})$        ▷ Eq. 7

**Output:** $\mathcal{V}$

---

### A.2.3 MODEL DETAILS

**Architecture.** We train the latent primitive diffusion model $g_\Phi$ using a Transformer-based architecture (Peebles & Xie, 2022) for scalability. Our final model (Eq. 9) is built with 28 layers with 16-

---

[1] https://github.com/facebookresearch/dinov2
[2] https://github.com/mlfoundations/open_clip

head attentions and 1152 hidden dimensions, leading to a total number of $\sim$1B parameters. Moreover, we employ the pre-normalization scheme (Xiong et al., 2020) for training stability. For noise scheduling, we use discrete 1,000 noise steps with a cosine scheduler during training. We opt for "v-prediction" (Salimans & Ho, 2022) with Classifier-Free Guidance (CFG) (Ho & Salimans, 2022) as the training objective for better conditional generation quality and faster convergence.

**Channel-wise Normalization.** Most importantly, given the distribution gap between the 3D coordinate $\mathbf{t}$ and the latent $E(\boldsymbol{X})$, one may carefully deal with the normalization of the input data to the diffusion model. Recall our diffusion target is a hybrid tensor $\mathcal{V} = \{\mathbf{t}, s, E(\boldsymbol{X})\}$, where $E(\boldsymbol{X})$ is the 3D latent in the KL-regularized VAE that is close to a Gaussian distribution. However, the 3D coordinate $\mathbf{t}$ is not normally distributed in the 3D space. This inter-channel distribution gap within the diffusion target will lead to suboptimal convergence if the data is globally normalized by a scalar (which is the common practice in 2D diffusion models[3]). Intuitively, our latent primitive diffusion model aims to solve a hybrid problem of point diffusion (Nichol et al., 2022) and latent diffusion (Rombach et al., 2022) simultaneously. To bridge this gap, we propose to normalize the input data in a channel-wise manner. Specifically, we trace channel-wise statistics (mean and standard deviation) over 50k random samples from the dataset. During the training phase, we keep them as constant normalizing factors and apply them to the input of the latent primitive diffusion model.

**Training.** We train $g_\Phi$ with a batch size of 1024 using an AdamW (Loshchilov & Hutter, 2017) optimizer. The learning rate is set to $1 \times 10^{-4}$ with a cosine learning rate warmup for 3k iterations. The probability of condition dropout for CFG is set to $p_0 = 0.1$. During training, we apply EMA (Exponential Moving Average) on the model's weight with a decay of 0.9999 for better training stability. The image-conditioned model is trained on 16 nodes of 8 A100 GPUs for 350k iterations, which takes around 14 days to converge. The text-conditioned model is trained on 16 nodes of 8 A100 GPUs for 200k iterations, which takes around 5 days to converge.

**VAE.** The 3D VAE for patch-wise primitive compression is built with 3D convolutional layers. We train the VAE on a subset of the entire dataset with 98k samples, finding it generalizes well on unseen data. The training takes 60k iterations with a batch size of 256 using an Adam (Kingma & Ba, 2014) optimizer with a learning rate of $1 \times 10^{-4}$. Note that, this batch size indicates the total number of PrimX samples per iteration. As our VAE operates on each primitive independently, the actual batch size would be $N \times 256$. We set the weight for KL regularization to $\lambda_{\mathrm{kl}} = 5 \times 10^{-4}$. The training is distributed on 8 nodes of 8 A100 GPUs, which takes about 18 hours.

**Inference.** By default, we evaluate our model with a 25-step DDIM (Song et al., 2020) sampler and CFG scale at 6. We find the optimal range of the DDIM sampling steps is $25 \sim 100$ while the CFG scale is $4 \sim 10$. The inference can be efficiently done on a single A100 GPU within 5 seconds.

### A.2.4 REVERSIBLE CONVERSION BETWEEN PRIMX AND GLB MESH

**Mesh to PrimX.** As introduced in the main paper (Sec. 3.1.2), we leverage a two-stage strategy to compute PrimX from a textured mesh. Given a textured mesh $F_{\mathcal{S}}$ that contains the shape, albedo, and material information, we convert it into PrimX with $N$ primitives via a good initialization followed by a rapid finetuning. Here, we introduce more details of this procedure in Algorithm 1. Our implementation to instantiate the volumetric sampling function of SDF that works for non-watertight mesh is derived from cuBVH[4].

**PrimX to Mesh.** As introduced in the main paper (Sec. 3.1.1), PrimX can be inversely converted back to a textured mesh in GLB format with minimal loss of information. The key is to utilize a high-resolution UV space for texturing instead of vertex coloring. We specify the details of this procedure in Algorithm 2, where we use xatlas[5] for UV unwrapping, nvdiffrast[6] for mesh-based rasterizer, and mcubes[7] for Marching Cubes (Lorensen & Cline, 1998).

---

[3] https://github.com/huggingface/diffusers/issues/437
[4] https://github.com/ashawkey/cubvh
[5] https://github.com/jpcy/xatlas
[6] https://github.com/NVlabs/nvdiffrast
[7] https://github.com/pmneila/PyMCubes

---

**Algorithm 2:** Extracting a Textured Mesh (GLB format) from PrimX

---

**Input** : PrimX $\mathcal{V} = \{\mathbf{t}_k, s_k, \mathbf{X}_k\}_{k \in [N]}$, Marching Cubes resolution $A$, chunk size $B$

$\{F_{\mathcal{V}}^{\text{SDF}}, F_{\mathcal{V}}^{\text{RGB}}, F_{\mathcal{V}}^{\text{Mat}}\} \leftarrow F_{\mathcal{V}}$

▷ *Shape Extraction*

$\{\mathbf{x}_i\}_{i \in [A^3]} \leftarrow$ Initialize a unit cube with a resolution of $A \times A \times A$

**for** $i \leftarrow 1$ **to** $A^3$ **do**

    **if** $\min_k ||\mathbf{x}_i - \{\mathbf{t}_k\}_{k \in [N]}||_2 > s_k$ **then**

        $F_{\mathcal{S}}^{\text{SDF}}(\mathbf{x}_i) \leftarrow \min_k ||\mathbf{x}_i - \{\mathbf{t}_k\}_{k \in [N]}||_2 \cdot \text{sign}(\mathbf{X}_k^{\text{SDF}})$     ▷ No query of PrimX

    **else**

        $F_{\mathcal{S}}^{\text{SDF}}(\mathbf{x}_i) \leftarrow F_{\mathcal{V}}^{\text{SDF}}(\mathbf{x}_i)$     ▷ Run in parallel with a chunk size $B$ in practice

$\{\mathbb{V}, \mathbb{F}\} \leftarrow$ Marching Cubes on the zero level set of $\{F_{\mathcal{S}}^{\text{SDF}}(\mathbf{x}_i)\}_{i \in [A^3]}$

▷ *Texture and Material Extraction*

Empty texture maps $(F_{\mathcal{S}}^{\text{RGB}}, F_{\mathcal{S}}^{\text{Mat}})$ and UV Mapping $\leftarrow$ UV unwrapping on $\{\mathbb{V}, \mathbb{F}\}$

$\{\mathbf{x}_i^{\text{uv}}\} \leftarrow$ Get validate sampling points in 3D with a rasterizer

$F_{\mathcal{S}}^{\text{RGB}}(\mathbf{x}_i^{\text{uv}}) \leftarrow F_{\mathcal{V}}^{\text{RGB}}(\mathbf{x}_i^{\text{uv}})$

$F_{\mathcal{S}}^{\text{Mat}}(\mathbf{x}_i^{\text{uv}}) \leftarrow F_{\mathcal{V}}^{\text{Mat}}(\mathbf{x}_i^{\text{uv}})$

$(F_{\mathcal{S}}^{\text{RGB}}, F_{\mathcal{S}}^{\text{Mat}}) \leftarrow$ inpainting with nearest neighbors based on UV mapping adjacency

$\mathcal{S} \leftarrow \{\mathbb{V}, \mathbb{F}, F_{\mathcal{S}}^{\text{RGB}}, F_{\mathcal{S}}^{\text{Mat}}, \text{UV Mapping}\}$     ▷ Packed in GLB format

**Output:** $\mathcal{S}$

---

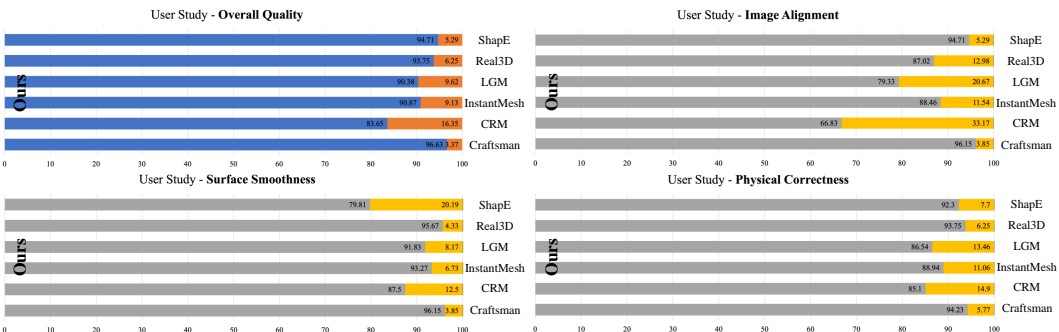

Figure 8: **User study.** We quantitatively evaluate comparison methods by conducting preference tests against our method on four dimensions. The results show that 3DTopia-XL has the highest preference rate compared with each of other methods.

### A.3 ADDITIONAL EXPERIMENTS

#### A.3.1 USER STUDY

We conduct an extensive user study to evaluate image-to-3D performance quantitatively. We opt for an output evaluation (Bylinskii et al., 2022) for user study, where each volunteer is shown with a pair of results comparing a random method against ours, and asked to choose the better one in four aspects: **1)** Overall Quality, **2)** Image Alignment, **3)** Surface Smoothness, and **4)** Physical Correctness. One of the samples presented to the attendees is shown in Figure 9. A total number of 48 paired samples are provided to 27 volunteers for the flip test. We summarize the average preference percentage across all four dimensions in Figure 8. 3DTopia-XL is the best one among all methods. Although the image alignment of our method is only a slight improvement against reconstruction-based methods like CRM, the superior quality of geometry and the ability to model physically based materials are the keys to producing the best overall quality in the final rendering.

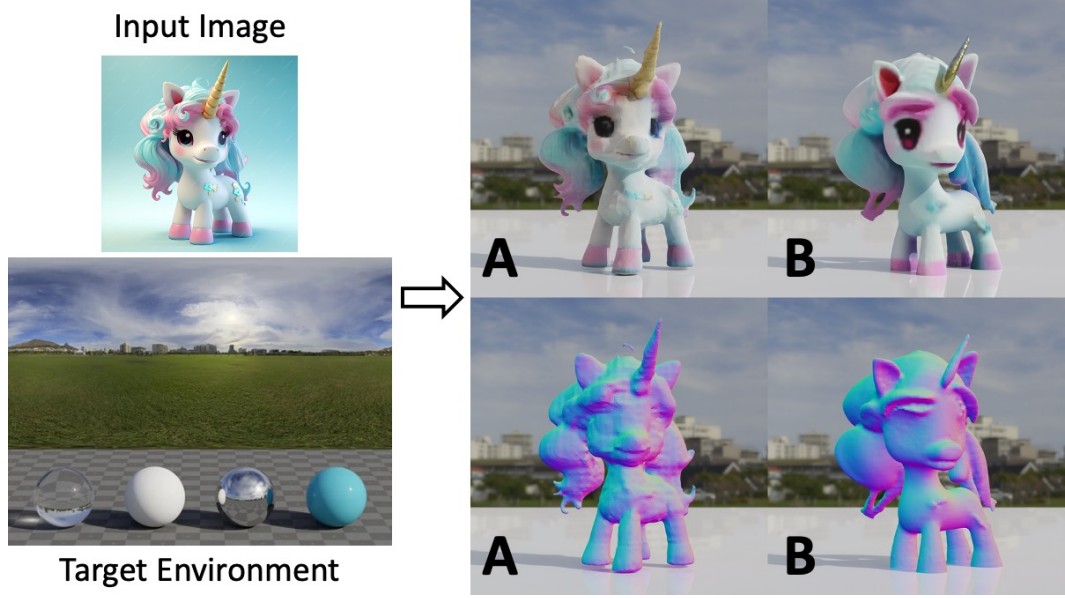

Figure 9: **User study sample.** For each sample in the user study, we present to the attendee with the input image (upper left) and target environment illuminations (bottom left) for rendering the mesh. Each volunteer is asked to choose the better one from A/B across four dimensions: 1) Overall quality, 2) Image alignment, 3) Surface smoothness, and 4) Physical correctness of renderings. The order and notation of methods are randomized and anonymized.

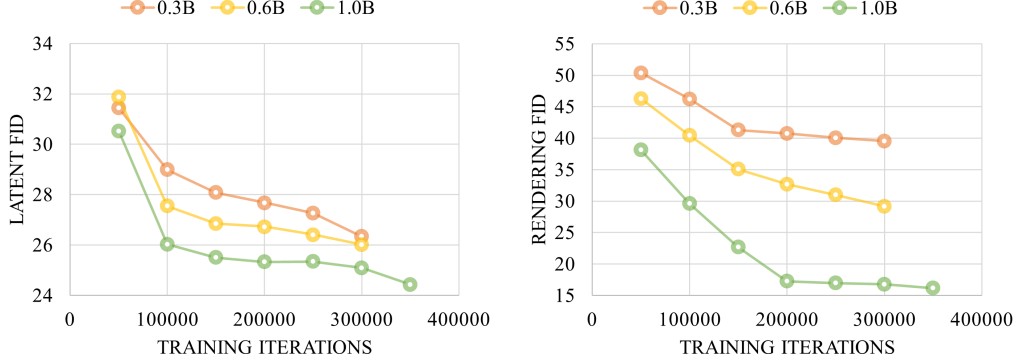

Figure 10: **Scaling up 3DTopia-XL improves FID.** As the computation and model size scale up, the model performance improves consistently. For metrics, we consider Latent-FID which is computed in the latent space of our VAE and Rendering-FID which is computed on the DINO (Oquab et al., 2023) embeddings extracted from images rendered with Eq. 6.

### A.3.2 SCALING

We further investigate the scaling law of 3DTopia-XL against model sizes and iterations. For metrics, we use Fréchet Inception Distance (FID) computed over 5k random samples without CFG guidance. Specifically, we consider Latent-FID which is computed in the latent space of our VAE and Rendering-FID which is computed on the DINO (Oquab et al., 2023) embeddings extracted from images rendered with Eq. 6. Figure 10 shows how Latent-FID and Rendering-FID change as the model size increases. We observe consistent improvements as the model becomes deeper and wider. Table 5 also demonstrates that longer sequence (smaller patches) leads to better performance, which may come from the findings in the vanilla DiT that increasing GFlops leads to better performance.

Table 5: **Longer sequence leads to better convergence.** Given a fixed PrimX parameter budget of 1.05M, we compare the models trained with $\{N = 256, a = 16\}$ and $\{N = 2048, a = 8\}$.

| Setting | Rendering-FID ↓ | Latent-FID ↓ |
|---|---|---|
| $N = 256$ | 76.31 | 104.8 |
| $N = 2048$ | 16.16 | 24.43 |

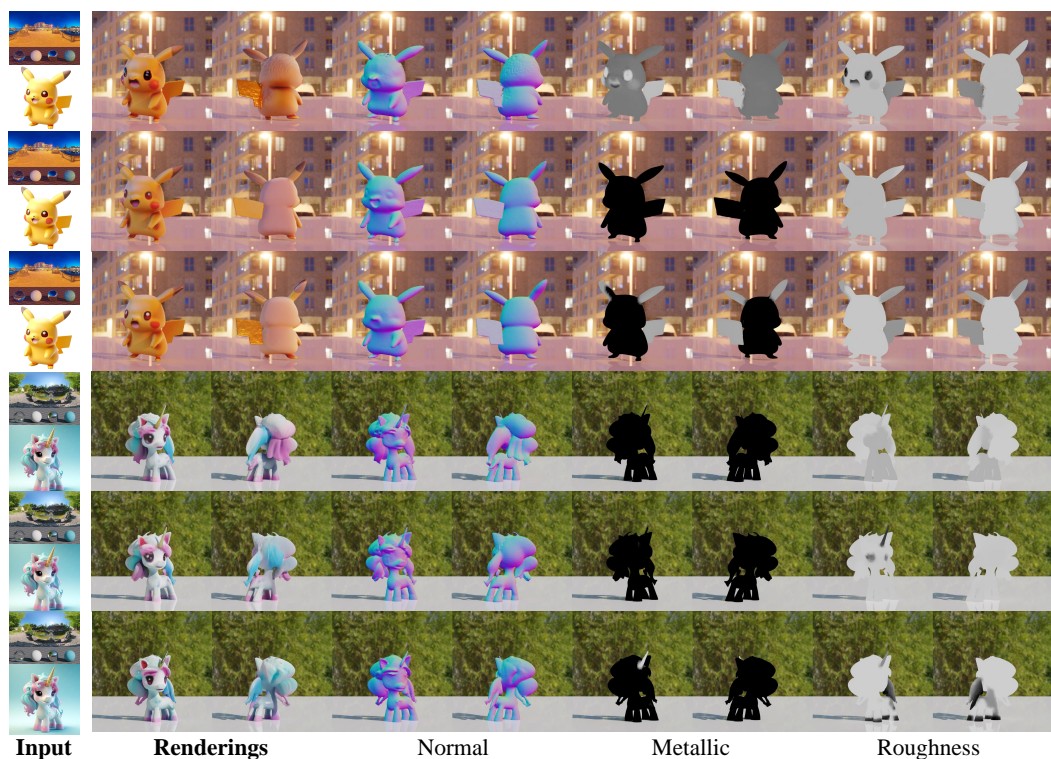

| **Input** | **Renderings** | Normal | Metallic | Roughness |
|---|---|---|---|---|

Figure 11: **Sampling diversity.** Given the same input image, 3DTopia-XL can generate diverse 3D assets by varying random seeds only. Zoom in for diverse shapes and spatially varied PBR materials.

### A.3.3 SAMPLING DIVERSITY

At last, we demonstrate the impressive sampling diversity of 3DTopia-XL as a generative model, as shown in Figure 11. Given the same input image and varying random seeds, our model can generate diverse high-quality 3D assets with different geometry and spatially varied PBR materials.

### A.3.4 ABLATION STUDY ON PRIMX INITIALIZATION

In this section, we conduct ablation studies on the impact of different initialization strategies for mesh to PrimX conversion (Algorithm 1). We compare three alternatives here:

- Uniform + Farthest (Ours): 1) we first perform uniform sampling to get $\hat{N}$ candidate points; and 2) we run farthest point sampling on the candidate point set to get $N$ primitives and initialize their scales to ensure coverage.

- Farthest: directly perform farthest point sampling to get $N$ primitives with a unique global scale factor as in M-SDF (Yariv et al., 2023).

- Coverage: 1) we first perform farthest point sampling to get $\frac{3}{4}N$ primitives; 2) a uniformly sampled point set is used to test the coverage by existing primitives, and points not covered are held out; and 3) we perform the second farthest point sampling on the held-out set to get the rest $\frac{1}{4}N$ primitive.

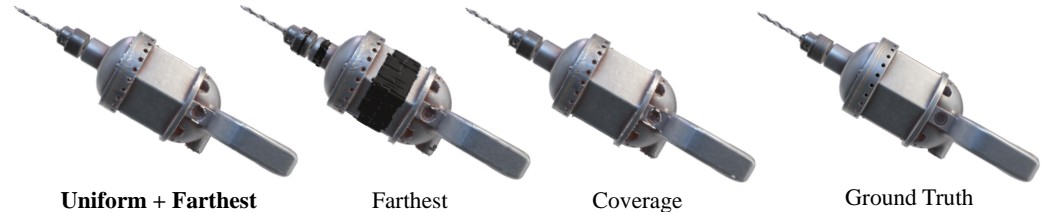

| Uniform + Farthest | Farthest | Coverage | Ground Truth |

Figure 12: **The impact of different initialization strategy for mesh to PrimX.**

Table 6: **Quantitative evaluations of different initialization strategies for mesh to PrimX.**

| Solution | PSNR-$F_{\mathcal{S}}^{\mathrm{SDF}}$ ↑ | PSNR-$F_{\mathcal{S}}^{\mathrm{RGB}}$ ↑ | PSNR-$F_{\mathcal{S}}^{\mathrm{Mat}}$ ↑ |
|---|---|---|---|
| **Uniform + Farthest** | 72.12 | 26.26 | 21.65 |
| Farthest | 56.86 | 14.30 | 10.16 |
| Coverage | 71.38 | 26.06 | 21.41 |

As shown in Figure 12, the "Farthest" solution is sensitive to the topology, which may lead to the insufficient number of primitive allocated to the flattened surface with a few mesh faces, causing the gap in the drill. Our final solution achieves comparable quality with the complicated "Coverage" solution and is capable of modeling fine-grained geometric details and consistent texture and material with ground truth. However, due to unnecessary computation overhead introduced by the latter solution, we choose the "Uniform + Farthest" initialization strategy as the final solution which is simple but effective. Quantitative results in Table 6 also confirm the above observation.

### A.3.5 More Results

We present more image-conditioned and text-conditioned generation results in Figure 7.

