# OpenReview forum: "3DTopia-XL: Scaling High-quality 3D Asset Generation via Primitive Diffusion"
_ICLR.cc/2025/Conference — ICLR 2025 Conference Withdrawn Submission_

### Official Review · Reviewer_XGY3 · 2024-10-24

**Soundness:** 2
**Presentation:** 3
**Contribution:** 2
**Rating:** 3
**Confidence:** 4

**Summary:**

The paper presents a method for 3D shape and PBR texture generation. The key to the method is the proposed representation PrimX — each object is represented by a set of primitive cubes on the surface, with each primitive containing position, scale, and volumetric SDF and PBR values, thus forming a tensorized representation. This representation can be converted to and from textured meshes, and compressed into a more compact latent space using a primitive-level VAE. A DiT is trained on this latent space for conditioned 3D generation. The authors conducted some comparisons and ablations to demonstrate the superiority of the method.

**Strengths:**

1. The writing is clear and easy to follow.
2. Presented a novel representation for 3D object generation.
3. The proposed method can also generate PBR.
4. Thorough ablations are done regarding the hyper-parameters of the proposed method.
5. The scaling result of the method shows its potential to further increase quality by increasing data and model size.

**Weaknesses:**

1. Discussion and comparison to sparse-voxel based generation methods are omitted, including but not limited to:
    1. Generalized Deep 3D Shape Prior via Part-Discretized Diffusion Process, CVPR 2023
    2. SDFusion: Multimodal 3D Shape Completion, Reconstruction, and Generation, CVPR 2023
    3. Locally Attentional SDF Diffusion for Controllable 3D Shape Generation, SIGGRAPH 2023
    4. One-2-3-45++: Fast Single Image to 3D Objects with Consistent Multi-View Generation and 3D Diffusion, CVPR 2024
    5. Make-A-Shape: a Ten-Million-scale 3D Shape Model, ICML 2024
    6. XCube: Large-Scale 3D Generative Modeling using Sparse Voxel Hierarchies, CVPR 2024
    7. MeshFormer: High-Quality Mesh Generation with 3D-Guided Reconstruction Model, NeurIPS 2024

    These methods are also parameter-efficient thanks to sparsity.
2. The shape and texture generation quality is lower than sota methods like CLAY, MeshLRM, and MeshFormer. The PBR also lacks high-frequency details.
3. Factual error in A.1.1: vecset can be extended to store color or PBR properties, by adding these properties along with PE in the encoding stage and decoding them by querying with positions. They are also differentiable renderable when combined with a differentiable marching cube algorithm and a differentiable rasterizer.
4. 4.2 Fig. 5 comparison with other methods: in the qualitative result part, all the meshes are not aligned, placed at different locations, and with different scales. Also, quantitative results on the image-to-3D task besides user studies are not reported.
5. Although claimed in the abstract, no real-world test cases are shown.

**Questions:**

1. The proposed method does not need differentiable rendering. It is also unsure how to train a latent 3D diffusion model using the so-called direct 2D supervision. That said, the claim of the possibility to learn from 2D images using PrimX is over-claimed.
2. How are the primitives ordered for a single object during the diffusion process? Will the order affect generation quality?
3. What is the advantage of using a local VAE instead of a global one, does it bring better quality? Or is it just cheaper to train? If you can train a DiT of ~1B, training a global VAE shouldn't be a big burden. The authors should provide this additional ablation on the VAE.
4. Although the authors claimed to be “rapidly tensorizable”, the speed for texture mesh to PrimX is actually very slow, as each shape takes 1.5 min for the conversion. I wonder how long it takes to process all these 250k training objects.
5. 4.1 representation evaluation: why not compare with sparse voxel and vecset? Also, what if the number of params of all methods is increased, like 5M and even 25M? Will the proposed representation still have privilege? The proposed representation also failed to reproduce texture details, e.g. the eyes in Fig 4. Can this problem be alleviated with more parameters? Besides, I think a more valuable comparison would be fixing the number of parameters in the encoded/latent space, as this is where the diffusion model runs.

Given the limited improvement over existing methods, the untenable motivation, and the missing evaluations, I lean to reject this paper. However, I'm willing to raise my score if the authors provide justifications for my above questions, and add missing results as listed in the weaknesses.

---

### Official Review · Reviewer_1Sfy · 2024-10-25

**Soundness:** 3
**Presentation:** 3
**Contribution:** 2
**Rating:** 5
**Confidence:** 4

**Summary:**

- This paper introduces a 3D Physically-Based Rendering (PBR) assets  generation method utilizing an expressive and efficient 3D representation, dubbed PrimX.
- The method leverages two main techniques:  Primitive Patch Compression and Latent Primitive Diffusion, effectively balancing generation speed with quality, achieving PBR asset creation within just 5 seconds of denoising.
- The proposed method consistently outperforms baseline methods in text-to-3D and image-to-3D settings.

**Strengths:**

**1.** The paper is well-motivated, effectively addressing the PBR rendering challenges for 3D asset generation.

**2.** The design of the model and its pre-processing pipeline (including "Mesh to PrimX", Prim Patch Encoder, and Primitive Diffusion) are well-conceived.

**3.**  The paper is well-written and provides extensive qualitative results to demonstrate its superiority (e.g., high geometric fidelity of PBR assets). The authors also include comprehensive ablation study and user study, which confirm the effect of each component and highlight the model's performance.

**Weaknesses:**

**1.** The primary concern is the $\textcolor{blue}{\text{apparent similarity}}$ between PrimX (the core design of this work) and PrimDiffusion (Chen et al., 2023b), which is scarcely mentioned, even in the "difference with related work"  ( **Sec.** A.1.1). The authors should clarify the fundamental differences between  two methods and prove (or clarify) why extending 3D **Primitives** to **PrimX** which including the PBR attribute (Material $\in \mathbb{R}^{a^{3}\times 2}$) is non-trivial.

**2.** Although the authors provide geometric and texture reconstruction results (e.g., **Fig.** 1, **Fig.** 5, and **Fig.** 7), they only show the predicted RGB images (texture) from the front view which could potentially be directly derived from the input image. The authors should showcase texture rendering from diverse viewpoints, including  side and  $\textcolor{blue}{\textbf{back views}}$, to better validate the model's capabilities.

**3.** Compared to cutting-edge 3D generation works (e.g., Real3D, LGM, CRM, InstantMesh), 3D Topia-XL requires substantial training resources and time, taking approximately **128** NVIDIA A100 GPUs and **14** days to converge.  Apart from the mentioned "v-prediction", could pre-trained 2D models or other training strategies help accelerate convergence? How can the authors reduce training costs to democratize the proposed 3D generative model?

**4.** In the Image-to-3D setting (**Sec.** 4.2), 3DTopia-XL achieves overwhelming superiority in PBR rendering due to baseline methods' incapability to support spatially varied materials. It may be beneficial for the authors to render objects without reflections in a fixed and simplified lighting environment, then quantitatively compare with other methods on a common GSO dataset and report PSNR, SSIM, LPIPS, and CD metrics.

If the authors could address my concerns by providing corresponding quantitative or qualitative results based on **the weaknesses** and **review feedback**, I will consider improving my score.

**Questions:**

- After generating the latent PrimX via latent primitive diffusion (**Fig.** 3), is it necessary to convert PrimX to mesh representations for the generation of 3D PBR assets in practical applications?
- In **Sec.** A.2.4,  the runtime of the "PrimX to Mesh" process (which includes UV unwrapping, Rasterizer, and Marching Cubes) is missed.

---

### Official Review · Reviewer_zK5B · 2024-10-31

**Soundness:** 3
**Presentation:** 2
**Contribution:** 3
**Rating:** 6
**Confidence:** 3

**Summary:**

3DTopia-XL is a high-quality 3D generative model designed to meet the growing demand for efficient 3D asset creation in fields such as gaming, virtual reality, and film. It introduces an innovative representation, PrimX, which encodes complex shapes, textures, and materials into a compact tensor format, supporting Physically Based Rendering (PBR) for realistic visuals. The model employs a Diffusion Transformer framework that facilitates efficient 3D asset generation from text or image inputs through its unique Primitive Patch Compression and Latent Primitive Diffusion techniques. Additionally, 3DTopia-XL includes optimized algorithms for extracting detailed PBR assets, ensuring easy integration into graphics engines. Experiments demonstrate that 3DTopia-XL significantly outperforms existing methods in producing high-resolution, finely detailed 3D assets, making it a promising foundation for advanced 3D generative modelling applications.

**Strengths:**

1. The paper introduces a novel tensor-based 3D representation called PrimX, which efficiently encodes geometry, albedo, and material properties. This representation allows for high-quality, physically-based rendering (PBR) assets with smooth geometry and intricate texture details, offering a more compact and efficient solution compared to traditional methods.

2. The proposed 3DTopia-XL model leverages a Diffusion Transformer to perform 3D generation from textual or visual inputs. This framework, combined with PrimX, enables high-quality and large-scale 3D asset generation, accommodating complex tasks like image-to-3D and text-to-3D conversion with better quality and efficiency compared to existing approaches.

3. The paper provides a detailed and efficient method for converting the PrimX representation back into a textured mesh in GLB format. This process ensures high-quality asset extraction by applying techniques like UV unwrapping, dilating, and inpainting, making the output ready for various downstream applications in graphics engines, with minimal additional processing required.

**Weaknesses:**

1. In Eq. 3, v is defined as a set of N volumetric primitives; however, it is not clear what each primitive denotes.

2. It’s not immediately clear what N and D represent. Are these dimensions related to the number of primitives and data channels (e.g., geometry and texture details), or is there another interpretation? Further defining these terms would add clarity.

3. The paper mentions using initialized positions and scales, where I is a unit local voxel grid. However, it’s not entirely clear how these initialized positions and scales contribute to the overall volumetric representation in the PrimX model. More context on what this setup accomplishes geometrically would help clarify this step.

4. Unclear about the role of permutation equivariance in PrimX and its impact on the Transformer model's design. Clarifying how permutation equivariance inherently maintains structure among primitives or why it aligns well with Transformer processing would be helpful.

**Questions:**

Please see the weaknesses.

---

### Official Review · Reviewer_8JMj · 2024-11-03

**Soundness:** 2
**Presentation:** 3
**Contribution:** 2
**Rating:** 5
**Confidence:** 4

**Summary:**

This paper introduces 3DTopia-XL, a 3D generative model designed to create 3D assets from both images and text. It extends 3D representation from M-SDF to support color and PBR materials, encoding shape, albedo, and material fields into a compact tensor format. The model framework, built on the Diffusion Transformer (DiT), incorporates Primitive Patch Compression and Latent Primitive Diffusion. Extensive experiments show that 3DTopia-XL effectively generates convincing 3D assets from both textual and visual inputs.

**Strengths:**

Strengths:
1）It extends the 3D representation from M-SDF to support color and PBR materials and integrates it into the generative framework, enabling the generation of 3D assets containing PBR materials from text and image conditions. This introduces a new approach to 3D representation for native 3D generative tasks.
2）Ablation studies on the number and resolution of primitives validated the selection of resolution.
3）The presentation of the paper is good.

**Weaknesses:**

Weakness：
1）My concern is regarding the accuracy of the PBR materials. The fourth row in Figure 7 shows noticeably incorrect PBR materials. The paper claims to generate 3D assets with PBR textures, so it should include experiments on a 3D synthetic dataset where the ground truth metallic and roughness data are available.  For example, quantitative evaluations can be handily conducted on a synthetic dataset with ground truth PBR data using  PSNR or SSIM for albedo, metallic, and roughness maps.
2）As mentioned in line 292, the paper introduces patch-based compression aimed at incorporating inter-channel correlations between geometry, color, and materials. However, there is no experimental evidence to support this. As shown in Table 3, comparing line 1 and line 2, when the grid compression rate remains the same, reducing the feature dimensions representing different geometry, color, and materials from 6 to 1 actually results in a decrease in PSNR.  Authors could provide additional experiments or analyses that directly demonstrate the benefits of incorporating these correlations, such as comparing the proposed approach to a baseline that processes geometry, color, and materials independently.
3）It seems that the experimental results are missing an ablation study where patch-based compression is not applied. It would be even better if qualitative comparison results could be added.  The author could include an ablation study that compares their patch-based compression approach to a baseline without compression, showing both quantitative metrics and qualitative visual comparisons.
4）The current generated results lack diversity. It is better to include quantitative measures of diverse objects with complex topologies, such as challenging chairs, plants, and buildings,

In summary, I have concerns about the novelty of the paper. The 3D representation primarily extends existing work on M-SDF, and the performance of the extended features, such as PBR material, is not satisfactory. Additionally, despite the use of patch-based compression, the reported computational cost is substantial, requiring 16 nodes of 8 A100 GPUs around 14 days to converge. In comparison, InstantMesh only needs 8 NVIDIA H800 GPUs. Moreover, compared with instant mesh, the generated geometric results in this paper tend to be overly smooth, losing geometric details, as shown in Figure 5.

**Questions:**

Questions：
See weakness 1, 2.

---

### Note · Authors · 2024-11-15

I have read and agree with the venue's withdrawal policy on behalf of myself and my co-authors.